# Factors Impacting Family Planning Use in Mali and Senegal

**DOI:** 10.3390/ijerph17124399

**Published:** 2020-06-19

**Authors:** Aissata Mahamadou Sidibe, Paul I Kadetz, Therese Hesketh

**Affiliations:** 1Center for Global Health, Zhejiang University, Hangzhou 310058, China; a.sidibe@zju.edu.cn (A.M.S.); paulkadetz@gmail.com (P.I.K.); 2The Institute for Global Health, University College London, London WC1N1EH, UK

**Keywords:** family planning, contraception, knowledge, attitudes and practices, Mali, Senegal

## Abstract

The total fertility rate in Mali (6.2) is the third highest in the world. Despite sociocultural similarities, the total fertility rate in neighboring Senegal is 4.2. The aim of this study is to identify factors which may help to explain the differences between the two countries and which may thereby inform family planning policy in Mali. A cross-sectional study was conducted with a convenience sample of 602 married women aged 16–50 from urban and rural sites in southern Mali and Senegal. A total of 298 respondents from Mali and 304 from Senegal completed a structured questionnaire between July and October 2018. In total, 11.1% of the Malian respondents and 30.9% of the Senegalese respondents were currently using family planning, and 34.6% and 40.5%, respectively, had ever used a modern family planning method. Pressure from husbands was cited as a primary influence for having more children (in 50.3% of Malians and 45.4% of Senegalese, *p* = 0.000). Women’s age, education level, and knowledge of different contraceptive methods were associated with ever use of contraceptives. After adjustment for confounders, discussing family planning with one’s husband was the strongest predictor of contraceptive use among both Senegalese (OR = 3.4, 95% CI (1.9–6.3), *p* = 0.000) and Malian respondents (OR = 7.3, (4.1–13.3), *p* = 0.000).

## 1. Introduction

Mali and Senegal are neighboring countries in West Africa with decades of shared history, dating to when they were collectively known as the Empire of Mali up to the point of their colonization by France in 1892 [1]. In 1959, the Sudanese Republic (Mali) and Senegal formed the Mali Federation, independent of France. However, in 1960, after only two months, Senegal withdrew from the Mali Federation and the Sudanese Republic was renamed Mali in September 1960. The union of the two countries was fully dissolved in 1989 [1]. Despite many sociocultural similarities in educational, legal, political, and healthcare sectors, as well as in the predominance of Islam and similar ethnic and age group demographics, the total fertility rate (TFR) in Mali is considerably higher than of Senegal [2].

Virtually unchanged for decades, the TFR in Mali has decreased only slightly from 7.1 in 1987 to 6.3 in 2018 [3]; it is now the third highest in the world and significantly higher than the average TFR of 4.8 across sub-Saharan Africa [4]. The modern contraceptive use rate among married women in Mali is 16% [3]. Mali’s population of 20 million is predicted to double by 2035 [5]. In neighboring Senegal, with a population of 16.6 million, the TFR decreased from 6.4 in 1987 to 4.2 in 2017, which is one of the lowest in sub-Saharan Africa. The prevalence of modern contraceptive use among married women in Senegal is 26% [6]. 

A major difference between the two countries is their security situation. Since 2012, the occupation of the northern regions of Mali by armed groups has plunged the country into conflict, which has worsened its already fragile health sector. In comparison, Senegal is considered to be one of the more politically stable countries in West Africa [7]. According to Tunçalp et al. (2015), significant strains were placed on the few reproductive services available in the south of Mali as more internally displaced women relocated there [8]. However, the health systems in both countries have been severely compromised by a long-term lack of investment [9]. In 2016, Mali had 0.52 and Senegal had 0.4 doctors, nurses, and midwives per 1000 of the population, well below the average of 1.5 per 1000 in sub-Saharan Africa and the WHO recommendation of 2.3 per 1000 [10]. The maternal mortality ratios in Mali and Senegal are among the highest in the world, with 325 deaths and 315 deaths per 100,000 live births, respectively [3,6]. The under-five mortality rate in Mali was estimated to be 101 per 1000 in 2018 [3], compared to 56 per thousand in Senegal [6]. The high under-five mortality rate in northern and central Mali is, at least partly, attributable to the very poor socioeconomic and security situations in these areas. 

This research adopts the World Health Organization’s definition of family planning as “the ability of individuals and couples to anticipate and attain their desired number of children and the spacing and timing of their births” [11]. Hence, “family” is framed here to be inclusive—even though all women in our sample were married—and “planning” is a broad category of awareness and capabilities needed to prepare for one’s future family that includes contraceptive methods to control pregnancy. In Mali, family planning (FP) is included in the minimum package of basic health services. The major public health facilities that offer family planning include community health centers, dispensaries, maternity clinics, and referral centers. Nurses and midwives are the main providers of FP services at the public health facilities. Physicians will usually only treat women who experience complications during pregnancy or those seeking tubal ligation [12]. In general, there is a lack of availability of all contraceptive methods. Facilities do not always have the resources to provide long-acting or permanent contraceptive methods in rural and urban areas. Contraceptives may also be provided by international and local Non Governmental Organizations, such as “Marie Stopes International”, who provide family planning in 70 community health centers in seven regions in Mali [13]. 

In Senegal, family planning is also included in the minimum package of basic health services. Public health facilities (hospitals, health centers, and clinics), as well as private facilities (hospitals and clinics) and pharmacies, offer reproductive health services including family planning. However, in rural areas women have far less access to family planning because of limited services [14,15].

The policies that address family planning also differ between the two countries. It is government policy in both Mali and Senegal that any woman who is seen by a midwife or a gynecologist should be introduced to family planning methods. In 2011, Mali and Senegal were among eight francophone West African countries who approved the “Ouagadougou Call to Action”—a commitment to take concrete actions to increase family planning use [16]. The Malian government subsequently adopted various initiatives to make FP services more accessible, including reducing the cost of contraceptives; offering long-term FP methods in health centers and via mobile services; “introducing contraceptive injectables in community outreach services, involving both the private sector and civil society in family planning services, educating religious leaders and parliamentarians, enlisting peer educators for family planning, and supporting social franchising” [17]. 

Whereas in Senegal, “the government purchased contraceptive products from the national budget, eliminated import duties for contraceptives, added FP products to the formal drug distribution system, harmonized FP product prices across the service delivery system, introduced measures to reduce contraceptive stockouts, and strengthened social marketing activities. Other initiatives include providing long-term FP methods at service delivery points and through mobile services, introducing injectables in community outreach services, extending FP services to communities in 56 districts, promoting peer education for youth, and removing the requirement for husbands to authorize their wives to receive FP services” [18].

Numerous studies have demonstrated the value of restricting family size for improving a number of health outcomes, including maternal and child mortality rates [19,20,21]. Several studies have shown that women’s attitudes towards family planning are influenced by specific factors, including traditional beliefs, religion, and their level of knowledge of contraceptive methods [22,23,24,25]. For example, a study of 30 sub-Saharan African countries concludes that women were less likely to seek family planning in settings where higher social value was placed on larger families [26]. Other key factors impacting contraceptive use on the African continent include the number of children desired, a woman’s age at marriage, and whether the husband is in a monogamous marriage [27,28,29,30,31]. 

Few population-based studies regarding family planning have been conducted in Mali or Senegal, and most studies in the region have focused on fertility rate, fertility preference, the prevalence of contraceptive use, and child mortality [32,33,34,35,36,37,38,39]. For example, some studies have identified that women with higher reproductive health knowledge in both countries were more likely to use contraceptives; conversely, lower contraceptive use was associated with less knowledge of family planning [35,36]. In Mali, family planning was also associated with women’s experiences with having children, finding a husband, and their experience of socialization [37]. In Senegal, the discontinuation of family planning was associated with women’s health problems, side effects from using contraceptives, infrequent sex, an absent husband, and wanting to become pregnant [38,39]. The research presented in this paper extends the understanding of factors influencing family planning in both Mali and Senegal beyond a woman’s knowledge of family planning. 

Given the marked differences in TFR between Mali and Senegal, the overall aim of this research is to gain an understanding of the factors impacting the use of modern family planning methods in both countries, with a view toward informing policy in Mali. Specifically, this research seeks to understand how women’s socio-demographic characterisitics, knowledge, and beliefs influence contraceptive use; how decisions about family planning are made; and how women access family planning in Mali and Senegal. 

## 2. Materials and Methods

### 2.1. Study Design

We conducted a quantitative cross-sectional study using a structured questionnaire.

### 2.2. Study Setting and Sample

Because of the ongoing unstable security situation in the northern part of Mali, data was collected in the southern part of both countries. The capital city of each country (Bamako in Mali and Dakar in Senegal) were chosen as the most economically developed areas, with a TFR lower than the national TFR. In addition, one rural and one urban site were chosen from two areas of each country with high fertility rates—Koulikoro and San in Mali and M’bour and Saint Louis in Senegal, respectively—for a total of ten sites. Convenience sampling was used across a range of settings: hospitals, clinics, pharmacies, markets, and workplaces in urban areas, and health centers in rural sites. The participants were women married for at least six months and aged between 15 and 50.

### 2.3. Data Collection

Data were collected between July and October 2018. The women were approached by the researcher (AS) who introduced the study and invited the women to complete a questionnaire. Verbal consent was obtained. Given the low rate of French literacy and the use of local dialects in both Mali and Senegal, the questionnaire was offered as either a self-administered tool in French or as a structured interview in Bambara or Wolof, the two dominant languages of Mali and Senegal, respectively.

The questionnaire was designed after a review of the literature and was piloted as a structured interview with 15 married women from Mali living in China. Changes to the original pilot questionnaire were made according to respondent suggestions, and it was piloted again with 10 married women in Mali and Senegal. The questionnaire included socio-demographic information and questions concerning the knowledge, beliefs, and practices around family planning. Women were encouraged to contribute written or verbal additional comments upon completion of the questionnaire.

### 2.4. Data Analysis

Descriptive analyses compared the socio-demographic characteristics, knowledge, beliefs, attitudes, and behaviors and practices of family planning in Mali and Senegal. Pearson’s Chi-squared was used to compare the influences of relevant factors on ever using family planning in Mali and Senegal. Binary logistic regression was used to analyze the predictors of ever using modern contraceptive methods after controlling for confounders. The data were analyzed using *SPSS 21* (IBM Corp, New York, NY, USA).

### 2.5. Ethical Approval

This study was approved by the ethics review committee of the School of Public Health of Zhejiang University. The Malian National Health Bureau supplied a letter of collaboration and support and the Senegalese Association of Family Well-Being approved the research in Senegal.

## 3. Results

### 3.1. Socio-Demographic Characteristics of the Sample 

Table 1 presents the socio-demographic characterisitics of the sample. In total, data from 298 women from Mali and 304 from Senegal were eligible for analysis. While most of the socio-demographic measures were significantly different between the two countries, the numerical differences were mostly small, indicating similarities between the two samples. In both countries, around half of the respondents had at least secondary education, around one third worked outside the home, and nearly half had grown up in rural areas. Notable differences between Mali and Senegal were that 163 (54.7%) of the Malian respondents had married under the age of 18, compared to 134 (44.1%) of the Senegalese (*p* = 0.031), and 152 (51%) of the Malians compared to 118 (38.8%) of the Senegalese (*p* = 0.003) had arranged marriages. 

### 3.2. Knowledge, Attitudes, and Beliefs 

Table 2 compares the basic knowledge, attitudes, and beliefs in relation to family planning in Mali and Senegal. The overwhelming majority of respondents in Mali and Senegal (78% and 86%, respectively) were aware of the existence of family planning, but overall Senegalese respondents were more knowledgeable, especially about different methods of family planning. For example, 34% of Malians could not name a modern contraceptive compared with 24% of Senegalese. Women from both samples generally thought that family planning was a good idea, though only about one third thought that their family and friends approved of family planning. A high proportion of women in both countries, 78% in Mali and 81% in Senegal, believed that women have too many children in general.

The single major source of pressure to have more children was believed to be one’s husband, reported by 150 (50.3%) respondents in Mali and 138 (45.4%) in Senegal. However, only a small minority felt that having more children would prevent one’s husband from engaging in polygamy or divorcing his wife, though the proportions were higher in Senegal than in Mali. 

Overall, 213 (71.5%) Malian respondents believed that women should not use contraceptives without their husband’s permission compared with 187 (61.5%) Senegalese (*p* = 0.006). Similarly, 153 (51.4%) Malian and 166 (54.6%) Senegalese believed that family planning should be discussed and decided jointly between a woman and her husband. A total of 74 (24.8%) and 104 (34.2%) Malian and Senegalese respondents, respectively, believed that their religion, predominantly Islam, is against family planning use (*p* = 0.001).

### 3.3. Preference for Family Size and Use of Family Planning 

Table 3 shows the preferences for family size and family planning practices: 39.5% of the Malian and 23% of the Senegalese respondents wanted more than five children, with only 4 (1.3%) and 15 (4.9%), respectively, wanting one or two children. Meanwhile, 105 (35.2%) Malian respondents and 76 (25.0%) Senegalese believed that their fertility was a result of “God’s will”. 

There was a significant difference in current contraceptive use; 33 (11.1%) used contraceptives in Mali and 94 (30.9%) used contraceptives in Senegal. The most common contraceptives used in Mali were implants, the contraceptive injection (i.e., medroxyprogesterone acetate), and the natural methods of withdrawal and safe period. In Senegal, the most popular contraceptive was the contraceptive injection, followed by implants, the contraceptive pill, and natural methods. Ever use of contraceptives showed a much narrower gap between the two countries, at 35% and 41% in Mali and Senegal, respectively. Over two thirds of the Malians, 202 (67.8%), and over half of the Senegalese, 159 (52.3%), reported that they never discussed family planning with their husband.

### 3.4. Associations with Ever Using Modern Contraceptives 

Table 4 illustrates the crude and adjusted odds ratios (OR) for associations between key independent variables and respondents ever using modern contraceptives in Mali and Senegal. After adjusting for age, women’s education, husband’s education, and residence during childhood, several significant associations were identified. 

Malian respondents who had ever used contraceptives were more likely to (a) have a higher education level (OR = 4.1, (1.6–10.2), *p* = 0.003); (b) have a husband with a higher education level (OR = 4.7, (1.7–12.4), *p* = 0.002); (c) engage in independent social activity (OR = 2.5, (1.4–4.3), *p* = 0.001); (d) have better knowledge of contraceptive methods (OR = 17.3, (4.5–66.3), *p* = 0.000); and (e) discuss family planning with their husband (OR = 7.3, (4.1–13.3), *p* = 0.000). 

Senegalese respondents who had ever used contraceptives were more likely to (a) have a higher education level (OR = 3.4, 95%CI (1.2–9.5) *p* = 0.018); (b) have spent their childhood in urban rather than rural areas; (OR = 4.2, (2.4–7.5), *p* = 0.000) (c) be employed outside of the home (OR = 2.3, (1.3–4.3), *p* = 0.005); (d) engage in independent social activity (OR = 1.9, (1.1–3.4), *p* = 0.020); (e) be more knowledgeable about contraceptive methods (OR = 5.4, (1.5–20.1), *p* = 0.011); (f) believe that their religion is not against family planning (OR = 2.4, (1.1–5.0), *p* = 0.021); and (g) discuss family planning with their husband (OR = 3.4, (1.9–6.3), *p* = 0.000).

In both countries, women aged between 16 and 30 were less likely to have used contraceptives. Respondents from both countries with a positive attitude toward contraceptives were more likely to use them.

### 3.5. Access to Family Planning 

A number of women in both Mali and Senegal freely commented about access to family planning. In rural areas in both countries, NGOs are a major source of family planning services. Visits to villages are supposed to occur monthly, but women said that these visits were often irregular, that there was little or no choice of contraceptives (usually only injection or the pill), and that in Mali these services were not free. Marked differences were reported in family planning services offered in Bamako and Dakar. Despite a regulation in Bamako requiring midwives to offer contraceptives at postpartum visits and on vaccination days, women reported that this rarely happened due to the high volume of patients. In rural and urban Mali, contraceptives are not free, and several women complained that they could not afford it. However, in urban Senegal contraceptives are free of charge and women and their husbands were observed to receive an introduction to family planning during pre- and post-natal check-ups.

## 4. Discussion

Our study aimed to identify the factors which may impact the use of family planning in married women from two neighboring Islamic nations with different fertility rates, among the highest and lowest in Sub-Saharan Africa. In particular, we sought to explore the potential for lessons from Senegal for Mali, where rapid population growth is a potential threat. We found a low likelihood of currently or ever using family planning in both countries; although Senegal demonstrated a higher use and more positive influencing factors in most areas. Our findings highlight a number of important factors which influence contraceptive use in both countries. After adjustment for confounders, these factors include the discussion of family planning with one’s husband, women’s educational background, knowledge of different contraceptive methods, and involvement in social activities. 

The strongest independent factor in both Mali and Senegal was discussion and joint decision-making with one’s husband. The majority of respondents identified their husbands as the main source of pressure to have more children, and believed a woman does not have the right to use a contraceptive without her husband’s permission, leading to the need for husbands’ involvement in family planning decisions. This finding is supported by studies in Nepal, Kenya, and Ethiopia [40,41,42], which found that women who discussed family planning with their partners were twice as likely to use contraceptives in Kenya and Ethiopia and ten times more likely in Nepal. The apparent contradiction in our study is that the majority of the respondents (67.8% in Mali and 52.3% in Senegal) did not actually discuss family planning with their husband. It is unclear whether this is because of women’s embarrassment or the simple avoidance of conflict around a sensitive topic. The pressure from husbands to have more children may reflect the value placed on having many children as a sign of wealth (and cultural capital) or as a symbol of the husband’s masculinity. For example, a study in Mali found that women’s husbands forbade them from using contraceptives because of the husband’s desire for more children and because the use of contraceptives was perceived to be an indication of a woman’s infidelity [32]. 

Second, higher levels of women’s education are associated with greater use of family planning. This association has been identified in myriad contexts. For example, in a study of contraceptive use among reproductive-age women in 17 sub-Saharan African countries, educated women were more likely to use contraceptives than uneducated women [43]. In a study in Ghana, any amount of education was associated with the increased use of family planning methods [44]. The education of men was also associated with increased family planning, as was found in our Malian sample. Similarly, a study among Jordanian women identified that 55% of women who, according to Islamic law, had considered the use of contraceptives forbidden, but whose husbands had a high level of education, were currently using a contraceptive method [45]. 

Third, knowledge and attitudes in relation to family planning were, as expected, significantly associated with contraceptive use and may help to explain the differences between the two countries. Nearly a quarter of Malians (22%) and 14% of Senegalese had not even heard of family planning. Familiarity with more types of contraceptives and positive attitudes towards family planning were both predictors of contraceptive use. This clearly shows the value of public education programs. 

The difference between knowledge in Senegal and Mali may also relate to the source of information. In Senegal, the source is more likely to be a health care worker who is highly knowledgeable and able to offer FP services. In Mali, women are more likely to learn about family planning from the media or from a friend. The proactive approach in Senegal, with the introduction of family planning for women (and their husbands, where possible) at pre- and post-natal consultations may offer guidance to Mali, where, at present, accessing services is haphazard. This is despite the aforementioned government policy in Mali that any woman who sees a midwife or a gynecologist should be introduced to family planning methods. The irony is that the failure to do so was observed to be due to the volume of patients, which is, at least partially, a result of the high birth rate. 

Fourth, participating in social activities with other women is independently associated with the use of contraceptive methods in both countries. This factor was highly significant after controlling for confounders, suggesting that socializing with other women may lead to discussions of family planning and empowerment in making family planning decisions. The relationship between social activities or social networks and family planning has been documented in several studies. In a study in Kenya, Kohler et al. found that social influence, which emphasizes normative influences on one’s behavior, is the dominant means by which social networks affect women’s contraceptive use [46]. In Cameroon, Valente et al. found that women were more likely to use contraceptives, in general, and specific contraceptives in particular, when their social network approved of and encouraged contraceptive use and/or emphasized particular methods [47]. 

Our findings also identify differences between the type of contraceptive methods currently used by Malian and Senegalese women. The most frequently used contraceptive method by women in Senegal is injections, followed by implants. However, in Mali the most frequently used method is the implant, followed by an injection; though it is not clear if this is an outcome of preference or availability, as limited choice was mentioned as an issue by some respondents. These findings are consistent with a study in Burkina Faso, where implants and injections were also the two most frequently used contraceptive methods, accounting for 88% of the modern contraceptive methods used among the sample [48]. Although respondents from Mali and Senegal were aware of male condoms, the general perception was that male condoms were primarily for unmarried couples to use in order to avoid HIV and unwanted pregnancies. 

Our findings also show a strong preference for a large family size in both countries, though Malian’s preference is somewhat greater, and is consistent with a stronger belief that women’s fertility is an outcome of “God’s will”, illustrating the more conservative society of Mali. In addition, the overwhelming majority of women in both countries wanted to space births, reflecting an understanding of the benefits of spacing for both mother and child.

### Limitations

This research has several limitations. First, this is a small sample size from the south of each country, and thereby greatly limits the generalizability of the findings. Second, selection bias is likely, because of the voluntary participation of respondents, although efforts were made to include participants across the socio-demographic range. Third, reliance on self-reporting may have led to a social desirability bias. Lastly, we did not ask about household income (partly because of questionable reliability), but this may of course affect access to family planning, especially in Mali, where some women volunteered that family planning was difficult to afford.

## 5. Conclusions

To ensure economic survival as well as the health and well-being of the population, the high population growth in Mali needs to be controlled. Senegal clearly provides lessons for Mali in improving the uptake of family planning. Despite differences between the countries and the more progressive government and culture in Senegal, Mali could benefit from modeling Senegal’s greater access to family planning services offered in antenatal and postnatal care—which supports husband involvement—as well as in immunization clinics. These activities may, in part, result from the Senegalese government’s investment in free family planning services and better access to contraceptives. Furthermore, according to our sample, the greater knowledge of family planning among Senegalese women signals the need for education campaigns that detail the benefits of limiting family size.

## Figures and Tables

**Table 1 ijerph-17-04399-t001:** Socio-demographic characteristics of respondents.

	Mali	Senegal	Pearson Chi (χ^2^)	*p* Value
	*n* = 298	*n* = 304		
**Age**			1.31	0.520
16–20	64 (21.5)	56 (18.4)		
21–30	139 (46.6)	140 (46.1)		
31–50	95 (31.9)	108 (35.5)		
**Religion**			5.70	0.017
Islam	275 (92.3)	294 (96.7)		
Others	23 (7.7)	10 (3.3)		
**Current residence**			1.18	0.278
Urban	242 (81.2)	236 (77.6)		
Rural	56 (18.8)	68 (22.4)		
**Women’s education**			9.05	0.011
No education	55 (18.5)	38 (12.5)		
Primary education	94 (31.5)	129 (42.4)		
Secondary and above	149 (50.0)	137 (45.1)		
**Husband’s education**			0.87	0.646
No education	51 (17.1)	61 (20.1)		
Primary education	119 (39.9)	118 (38.8)		
Secondary and above	128 (43.0)	125 (41.1)		
**Employment outside the home**			1.91	0.167
Yes	109 (36.6)	95 (31.3)		
No	189 (63.4)	209 (68.8)		
**Age at marriage**			6.94	0.031
< = 18	163 (54.7)	134 (44.1)		
19–25	115 (38.6)	142 (46.7)		
> = 26	20 (6.7)	28 (9.2)		
**Marriage status**			4.69	0.030
Monogamous	218 (73.2)	245 (80.6)		
Polygamous	80 (26.8)	59 (19.4)		
**Childhood residence**			7.53	0.006
Urban	185 (62.1)	155 (51.0)		
Rural	113 (37.9)	149 (49.0)		
**Arranged marriage**			9.04	0.003
Yes	152 (51.0)	118 (38.8)		
No	146 (49.0)	186 (61.2)		

**Table 2 ijerph-17-04399-t002:** Knowledge, attitudes, and beliefs about family planning.

	Mali	Senegal	Pearson Chi (χ^2^)	*p* Value
	*n* = 298	*n* = 304		
**Heard about family planning**			7.01	0.008
Yes	232 (77.9)	262 (85.9)		
No	66 (22.1)	42 (13.8)		
**Can name how many modern contraceptive methods**			14.02	0.001
0	102 (34.2)	72 (23.7)		
1–3	148 (49.7)	150 (49.3)		
4+	48 (16.1)	82 (27.0)		
**Can name traditional contraceptive methods**			5.01	0.025
Yes	120 (40.3)	150 (49.3)		
No	178 (59.7)	154 (50.7)		
**Knowledge of modern contraceptives before marriage**			0.61	0.434
Yes	70 (35.7)	91 (39.4)		
No	126 (64.3)	140 (60.6)		
**What do you think of family planning**	*n* = 232	*n* = 262	11.00	0.004
Good	171 (73.7)	174 (66.4)		
Bad	32 (13.8)	66 (25.2)		
No opinion	29 (12.5)	22 (8.4)		
**What do your social networks (friends and family) think about family planning?**	*n* = 232	*n* = 262	1.83	0.400
Good	78 (33.6)	91 (34.9)		
Bad	100 (43.1)	122 (46.7)		
I don’t know their opinion	54 (23.3)	48 (18.4)		
**Having many children prevents a man from divorcing his wife**			9.07	0.011
Yes	42 (14.1)	72 (23.7)		
No	245 (82.2)	223 (73.4)		
Don’t know	11 (3.7)	9 (3.0)		
**Having many children prevents a man from taking another wife**			7.20	0.027
Yes	21 (7.1)	35 (11.5)		
No	266 (89.3)	248 (81.6)		
Don’t know	11 (3.7)	21 (6.9)		
**What are sources of pressure to have more children?**			28.13	0.000
Society/culture	19 (6.4)	22 (7.2)		
Family	22 (7.4)	38 (12.5)		
Husband	150 (50.3)	138 (45.4)		
Neighborhood/community	10 (3.4)	11 (3.6)		
Inheritance	15 (5.1)	41 (13.5)		
Women’s own desire	46 (15.4)	41 (13.5)		
Don’t know	36 (36.1)	13 (4.3)		
**Is your religion against family planning**			15.98	0.001
Totally against	74 (24.8)	104 (34.2)		
Not against	91 (30.6)	56 (18.4)		
Not against when for health reasons	62 (20.8)	80 (26.3)		
Don’t know	71 (23.8)	64 (21.1)		
**Does a woman have a right to use a contraceptive without her husband’s permission**			10.20	0.006
Yes	61 (20.5)	97 (31.9)		
No	213 (71.5)	187 (61.5)		
Don’t know	24 (8.1)	20 (6.6)		
**Who do you think should decide family planning in your family**			1.072	0.784
Woman	72 (24.2)	72 (23.7)		
Husband	61 (20.5)	53 (17.3)		
Couple	153 (51.4)	166 (54.6)		
Don’t know	12 (4.0)	13 (4.3)		

**Table 3 ijerph-17-04399-t003:** Family size preference and family planning use.

	Mali	Senegal	Pearson Chi (χ^2^)	*p* Value
	*n* = 298	*n* = 304		
**How many children do you want?**			48.86	0.000
1–2	4 (1.3)	15 (4.9)		
3–4	61 (20.5)	128 (42.1)		
5–6	66 (22.1)	34 (11.2)		
More than 6	52 (17.4)	36 (11.8)		
God’s will	105 (35.2)	76 (25.0)		
Don’t know	10 (3.4)	15 (4.9)		
**Number of children desired before marriage**			19.11	0.002
1–2	6 (2.0)	23 (7.6)		
3–4	39 (13.1)	61 (20.1)		
5–6	21 (7.0)	11 (3.6)		
More than 6	23 (7.7)	20 (6.6)		
God’s will	23 (7.7)	20 (6.6)		
Don’t know	186 (62.4)	169 (55.6)		
**Desire to space births**			1.24	0.537
Yes	274 (91.9)	277 (91.1)		
No	22 (7.4)	22 (7.2)		
Don’t know	2 (0.7)	5 (1.6)		
**Spacing years desired**	*n* = 274	*n* = 277	17.19	0.000
1–2 years	113 (41.2)	69 (24.9)		
3–4 years	145 (52.9)	182 (65.7)		
5+ years	16 (5.8)	26 (9.4)		
**Current number of children**			7.06	0.030
0–3	191 (64.1)	225 (74.0)		
4–6	76 (25.5)	58 (19.1)		
7+	31 (10.4)	21 (6.9)		
**Ever used a modern contraceptive**			2.23	0.135
Yes	103 (34.6)	123 (40.5)		
No	195 (65.4)	181 (59.5)		
**Current use of modern contraceptive**			35.61	0.000
Yes	33 (11.1)	94 (30.9)		
No	265 (88.9)	210 (69.1)		
**Current modern contraceptive methods used**	*n* = 33	*n* = 94	11.32	0.026
IUD	2 (6.1)	6 (6.5)		
Implant	15 (45.5)	17 (18.3)		
Injection	11 (33.3)	57 (60.6)		
Oral contraceptive	4 (12.1)	13 (14.0)		
Morning-after pill	1 (3.0)	1 (1.1)		
**Discussed family planning with husband**			15.03	0.000
Yes	96 (32.2)	145 (47.7)		
No	202 (67.8)	159 (52.3)		

**Table 4 ijerph-17-04399-t004:** Relationships between key variables and ever using contraceptives in Mali and Senegal.

Ever Using Modern Contraceptives												
	Mali						Senegal					
	Total	N (%)	Crude OR (95%)	*p* Value	Adjusted OR (95%)	*p* Value	Total	N (%)	Crude OR (95%)	*p* Value	Adjusted OR (95%)	*p* Value
**Age**												
31–50	95	42 (44.2)	1		1		108	59 (54.6)	1		1	
21–30	139	49 (35.3)	0.7 (0.4–1.2)	0.168	0.6 (0.3–1.1)	0.077	140	56 (40.0)	0.6 (0.3–0.9)	0.022	0.3 (0.2–0.6)	0.000
16–20	64	12 (18.8)	0.3 (0.1–0.6)	0.001	0.2 (0.1–0.5)	0.000	56	8 (14.3)	0.1 (0.1–0.3)	0.000	0.1 (0–0.3)	0.000
**Women’s education**												
No education	55	9 (16.4)	1		1		38	7 (18.4)	1		1	
Primary education	94	26 (27.7)	2.0 (0.8–4.6)	0.120	2.1 (0.9–5.2)	0.100	129	50 (38.8)	2.8 (1.1–6.8)	0.024	3 (1.1–7.9)	0.026
Secondary+	149	68 (45.6)	4.3 (2.0–9.4)	0.000	4.1 (1.6–10.2)	0.003	137	66 (48.2)	4.1 (1.7–10.0)	0.002	3.4 (1.2–9.5)	0.018
**Husband’s education**												
No education	51	6 (11.8)	1		1		61	17 (27.9)	1		1	
Primary education	119	34 (28.6)	3.0 (1.1–7.7)	0.022	2.4 (0.9–6.3)	0.075	118	43 (36.4)	1.5 (0.8–2.9)	0.251	1.3 (0.6–2.7)	0.553
Secondary+	128	63 (49.2)	7.3 (2.9–18.2)	0.000	4.7 (1.7–12.4)	0.002	125	63 (50.4)	2.6 (1.4–5.1)	0.004	1.5 (0.7–3.3)	0.285
**Childhood**												
Rural	113	33 (29.2)	1		1		149	34 (22.8)	1		1	
Urban	185	70 (37.8)	1.5 (0.9–2.4)	0.129	0.8 (0.4–1.4)	0.411	155	89 (57.4)	4.6 (2.8–7.5)	0.000	4.2 (2.4–7.5)	0.000
**Work**												
No	189	57 (30.2)	1		1		209	62 (29.7)	1		1	
Yes	109	46 (42.2)	1.7 (1.0–2.8)	0.036	1.5 (0.9–2.6)	0.142	95	61 (64.2)	4.3 (2.5–7.1)	0.000	2.3 (1.3–4.3)	0.005
**Marriage age**												
> = 26	20	15 (75.0)	1		1		28	16 (57.1)	1		1	
19–25	115	48 (41.7)	0.2 (0.1–0.7)	0.009	0.3 (0.1–1.1)	0.022	142	66 (46.5)	0.7 (0.3–1.5)	0.304	2.4 (0.9–6.4)	0.086
< = 18	163	40 (24.5)	0.1 (0–0.3)	0.000	0.3 (0.1–0.8)	0.071	134	41 (30.6)	0.3 (0.1–0.7)	0.009	1.8 (0.6–4.8)	0.272
**Arranged marriage**												
No	146	67 (45.9)	1		1		186	84 (45.2)	1		1	
Yes	152	36 (23.7)	0.4 (0.2–0.6)	0.000	0.6 (0.3–1.0)	0.069	118	39 (33.1)	0.6 (0.4–1.0)	0.037	1.3 (0.7–2.4)	0.489
**Has independent social activity**												
No	164	36 (22.0)	1		1		135	34 (25.2)	1		1	
Yes	134	67 (50.0)	3.6 (2.2–5.9)	0.000	2.5 (1.4–4.3)	0.001	169	89 (52.7)	3.3 (2.0–5.4)	0.000	1.9 (1.1–3.4)	0.020
**Modern contraceptive knowledge**												
1	25	4 (16.0)	1		1		19	4 (21.1)	1		1	
2–3	123	58 (47.2)	4.7 (1.5–14.4)	0.007	4.5 (1.4–14.3)	0.011	131	63 (48.1)	3.5 (1.1–11.0)	0.035	2.2 (0.6–7.9)	0.215
4+	48	39 (81.3)	22.8 (6.3–82.8)	0.000	17.3 (4.5–66.3)	0.000	82	56 (68.3)	8.1 (2.4–26.7)	0.001	5.4 (1.5–20.1)	0.011
**Religion is against family planning**												
Totally against	74	29 (39.2)	1		1		72	43 (41.3)	1		1	
Not against	91	47 (51.6)	1.7 (0.9–3.1)	0.111	1.6 (0.8–3.3)	0.150	84	37 (66.1)	2.8 (1.4–5.4)	0.003	2.4 (1.1–5.0)	0.021
Not against for health	62	40 (64.5)	0.9 (0.4–1.7)	0.657	1.3 (0.6–2.8)	0.546	57	35 (43.8)	1.1 (0.6–2.0)	0.744	1.6 (0.8–3.0)	0.200
**Women opinion about family planning**												
Good	171	90 (52.6)	1		1		174	99 (56.9)	1		1	
Bad	32	8 (25)	0.3 (0.1–0.7)	0.006	0.3 (0.1–0.8)	0.015	66	22 (33.3)	0.4 (0.2–0.7)	0.001	0.3 (0.2–0.6)	0.001
No opinion	29	3 (10)	0.1 (00–0.4)	0.000	0.1 (0–0.5)	0.003	22	2 (9.1)	0.1 (0–0.3)	0.001	0.1 (0–0.6)	0.008
**Partner discussion**												
No	202	25 (12.4)	1		1		159	29 (18.2)	1		1	
Yes	96	78 (81.3)	8.9 (5.1–15.5)	0.000	7.3 (4.1–13.3)	0.000	145	94 (64.8)	4.3 (2.5–7.3)	0.000	3.4 (1.9–6.3)	0.000

Note: Adjusted for age, women’s education, husbands’ education, and residence during childhood.

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
