# Peer review of "Factors Impacting Family Planning Use in Mali and Senegal"

_ijerph, 2020, doi:10.3390/ijerph17124399_

Round 1

Reviewer 1 Report

Contraceptive use is an important factor in demographic transition and is also a health behavior. The relative low prevalence of contraceptive use is an important family planning issue and debating agenda in Sub-Saharan Africa. This study tells the situation and related factors of contraceptive use in Mali and Senegal by information collected from a convenience sample. The findings are informative. However there are some limitations in the manuscript for providing a clear picture and to better communicate.

1 The use of contraception is related to whether fertility goal is reached, however in the analysis the gap between number of children already have and desired number of children is not clearly involved, which should be included in the multivariate analysis.

2 Both injections and implant have to be provided by professionals, but there is no mention on the service supply side, such as contraceptive service availability and accessibility.

3 Is there any similar study for women either in Mali or in Senegal? What is the unique contribution of this study to the existing knowledge about contraceptive use in the areas? The authors should address these issues in introduction or in discussion.

Author Response

Response to Reviewer 1 Comments

We very much appreciate the constructive comments of the reviewer. We have responded point-by-point below.

Point 1: The use of contraception is related to whether fertility goal is reached, however in the analysis the gap between number of children already have and desired number of children is not clearly involved, which should be included in the multivariate analysis.

Response 1: In our questionnaire, we didn’t ask women about the number of children they plan to have in the future, so we cannot determine if their fertility goal was reached or not. In addition, 35% of our Malian sample and 25% of the Senegalese sample reported that the number of children they have depends on 'God's will'. In other words, for a proportion of the sample individual desire for the number of children is not relevant.

Point 2: Both injections and implant have to be provided by professionals, but there is no mention on the service supply side, such as contraceptive service availability and accessibility.

Response 2: Contraceptive service availability and accessibility was added to the introduction section page 2, lines 53 to 64.

Point 3: Is there any similar study for women either in Mali or in Senegal? What is the unique contribution of this study to the existing knowledge about contraceptive use in the areas? The authors should address these issues in introduction or in discussion.

Response 3: Similar studies were added to the introduction section page 2 Line 90 to 96. The unique contribution of the study compared to the existing knowledge was added to page 2, lines 96 to 97.

Reviewer 2 Report

My discipline is in the arts and humanities, so I judged this essay on the clarity of the presentation, the language, and my ability to understand what the study was, why it was conducted, and what significance could be taken away from the results. The authors made very clear what the limitations of their study were, but still had a large enough control group that they were able to make conclusions and recommendations. 

Author Response

We very much appreciate the constructive comments of the reviewer. 

Reviewer 3 Report

Introduction

The authors's starting point are that Mali and Senegal are socioculturally similar.  I have no reason to doubt this, but I would like the authors to flesh out this argument a bit more.  I am always skeptical about claims that two neighboring countries are similar.  I would be more satisfied with more discussion of the context and history.  I would also like more background on health policy development in the two countries.

Why did you choose the terminology "family planning" over SRHR?

Design

The study design section focusses on the survey, but in the introduction there was mention of interviews with health authority informants. If this was part of a large study, it should be mentioned in the materials and methods section.

Results

What are "traditional" family planning methods?

Discussion

If the 'strongest independent factor' is decision making with one's husband, then the discussion should lead with that,  rather than with "God's will"

I'm not keen on the phrasing "more traditional culture in Mali" because it can sound a bit pejorative. 

Limitations

The authors note that the research comes from the South, not the North; but I would like more of a discussion on the generalizability.  In the introduction the authors note that a major difference - which affects health - between the two countries is the security situation in the north; but to what extent does this affect family planning services in the south?

Author Response

Response to Reviewer 3 Comments

We very much appreciate the constructive comments of the reviewer. We have responded point-by-point below.

Point 1: The authors’ starting point are that Mali and Senegal are sociocultural similar.  I have no reason to doubt this, but I would like the authors to flesh out this argument a bit more.  I am always skeptical about claims that two neighboring countries are similar.  I would be more satisfied with more discussion of the context and history.  I would also like more background on health policy development in the two countries.

Response 1: Background information concerning the context and history of the two countries, as well as the policy development of family planning in both countries was added as requested. Please refer to Page 1  lines 26-32 for the history and context of the two countries; and to Page 2 Line 65 to 80 for pertinent health policy development.

Point 2: Why did you choose the terminology "family planning" over SRHR?

Response 2: We chose the terminology family planning because SRHR is a broader category and family planning is considered a smaller part of SRHR.

Design

Point 3: The study design section focusses on the survey, but in the introduction there was mention of interviews with health authority informants. If this was part of a large study, it should be mentioned in the materials and methods section.

Response 3:   These were informal interviews carried-out during the project design phase, and as they were not part of the substantive project, we have removed this from the manuscript.

Results

Point 4: What are "traditional" family planning methods?

Response 4: The word 'traditional' was changed to 'natural' methods and explained on page 6, line 171.

Discussion

Point 5: If the 'strongest independent factor' is decision making with one's husband, then the discussion should lead with that, rather than with "God's will"

Response 5: Discussion order has been changed accordingly, and moved to page 13 lines 285-289.

Point 6: I'm not keen on the phrasing "more traditional culture in Mali" because it can sound a bit pejorative.

Response 6: 'Traditional culture' was changed to “more conservative society” on page 13 line 287.

Limitations

Point 7: The authors note that the research comes from the South, not the North; but I would like more of a discussion on the generalizability.  In the introduction the authors note that a major difference - which affects health - between the two countries is the security situation in the north; but to what extent does this affect family planning services in the south?

Response 7: Potential effects of the insecurity in the north on family planning services in the south is  explained in terms of the impact that internally displaced women from the north have on family planning.

Round 2

Reviewer 1 Report

The revision and explanation are satisfactory. No further comment.

Author Response

Authors Response

We thank the reviewer for their helpful comments. In response, between lines 53 and 57 on page 2, we include a definition of family planning (The WHO's 2008 definition) adopted for this paper and also differentiate FP from contraceptive use.